# Efficient Rectified Flow for Image Fusion

**Zirui Wang**[1]**, Jiayi Zhang**[2]**, Tianwei Guan**[3]**, Yuhan Zhou**[2]**,**
**Xingyuan Li**[4]**, Minjing Dong**[1]**, Jinyuan Liu**[2]*
[1] City University of Hong Kong [2] Dalian University of Technology
[3] Chinese University of Hong Kong [4] Zhejiang University
`zrwang23-c@my.cityu.edu.hk  xingyuan_lxy@163.com`

## Abstract

Image fusion is a fundamental and important task in computer vision, aiming to combine complementary information from different modalities to fuse images. In recent years, diffusion models have made significant developments in the field of image fusion. However, diffusion models often require complex computations and redundant inference time, which reduces the applicability of these methods. To address this issue, we propose RFfusion, an efficient one-step diffusion model for image fusion based on Rectified Flow. We incorporate Rectified Flow into the image fusion task to straighten the sampling path in the diffusion model, achieving one-step sampling without the need for additional training, while still maintaining high-quality fusion results. Furthermore, we propose a task-specific Variational Autoencoder (VAE) architecture tailored for image fusion, where the fusion operation is embedded within the latent space to further reduce computational complexity. To address the inherent discrepancy between conventional reconstruction-oriented VAE objectives and the requirements of image fusion, we introduce a two-stage training strategy. This approach facilitates the effective learning and integration of complementary information from multi-modal source images, thereby enabling the model to retain fine-grained structural details while significantly enhancing inference efficiency. Extensive experiments demonstrate that our method outperforms other state-of-the-art methods in terms of both inference speed and fusion quality. Code is available at `https://github.com/zirui0625/RFfusion`.

## 1  Introduction

In computer vision, image fusion is an important task aimed at merging two images from different modalities to obtain a fused image that contains complementary information from both modalities. Image fusion has wide applications across various scenarios. Infrared and visible image fusion (IVIF) [1, 2, 3, 4, 5] aims to enhance perception under adverse conditions by integrating the detailed information from visible images with the thermal radiation characteristics of infrared images. Medical image fusion (MIF) [6, 7] focuses on mitigating the information discrepancies between MRI and CT modalities to provide more comprehensive and accurate diagnostic support. Multi-exposure image fusion (MEF) [8, 9] and multi-focus image fusion (MFF) [10, 11] focus on merging images with different exposures and different focal planes, to synthesize high-quality photographic images.

In recent years, with the advent of Denoising Diffusion Probabilistic Models (DDPMs) [12], diffusion-based methods have been widely adopted across various computer vision tasks, including image fusion. DDPMs learn the denoising process from noisy observations back to clean images over the data distribution, thereby acquiring the ability to generate high-quality images.

---

*Corresponding author.

39th Conference on Neural Information Processing Systems (NeurIPS 2025).

Compared to traditional fusion approaches [13, 14], diffusion-based methods [15, 16, 17, 18] not only effectively integrate information from multiple source images but also significantly enhance the visual quality of the fused results. Benefiting from the powerful priors encoded in pre-trained diffusion models, these approaches demonstrate great potential in multi-task fusion scenarios, where a single unified framework can be adapted to various fusion tasks with remarkable performance.

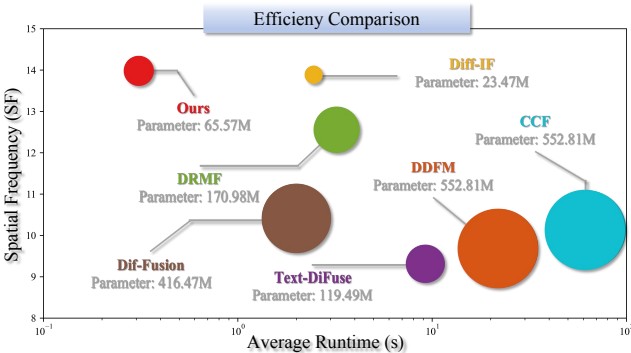

Figure 1: Efficiency comparison with the state-of-the-art diffusion-based methods.

Though diffusion-based methods have achieved remarkable progress in image fusion tasks, their long inference times pose significant challenges for real-world applications. Recently proposed approaches such as DDFM [15] and CCF [16] introduce fusion priors into the sampling process of diffusion models, effectively improving fusion quality. However, these methods typically require hundreds of sampling steps to achieve satisfactory results. Reducing the number of steps to improve efficiency often leads to a substantial drop in fusion performance. To accelerate inference in diffusion models, several strategies such as distillation and latent space diffusion have been widely explored. Nonetheless, their application to image fusion remains limited. **First**, existing distillation methods can enable single-step sampling but usually require fine-tuning tailored to specific model architectures and datasets, lacking generalization across diverse fusion tasks. **Second**, while latent space diffusion methods based on Variational Autoencoder (VAE) can significantly reduce computational costs, their training objective primarily targets image reconstruction rather than image fusion, leading to considerable challenges when applied directly to fusion scenarios. **Therefore, it is imperative to develop a sampling acceleration method that is tailored to image fusion tasks, capable of preserving fusion quality while maintaining generalizability.**

To address these challenges, we propose a novel method named **RFfusion**, which introduces the Rectified Flow mechanism into image fusion tasks for the first time. RFfusion significantly accelerates the inference process of diffusion models without requiring additional training and exhibits strong generalization across multiple fusion tasks. Specifically, we leverage Rectified Flow to construct a linear trajectory between the input images and the target fused image, embedding prior knowledge of the fused image during the sampling process to achieve efficient and high-quality single-step inference. Moreover, we incorporate the sampling process into the latent space and propose a two-stage training strategy to address the objective mismatch between VAE reconstruction and fusion tasks. In the first stage, we fine-tune the VAE to better capture critical features needed for image fusion. In the second stage, the optimized VAE is integrated into the overall fusion framework for joint training, further enhancing the model's adaptability to fusion scenarios. Extensive experimental results demonstrate that RFfusion not only substantially reduces the number of inference steps and improves computational efficiency, but also outperforms existing state-of-the-art methods across multiple standard image fusion benchmarks. Our contributions can be summarized as:

- We propose a novel Efficient Rectified Flow image fusion (RFfusion) framework that enables one-step sampling across various fusion tasks without requiring additional training, significantly reducing computational cost and inference time while achieving high-quality fused images.

- We introduce the image fusion task into the latent space to effectively reduce computational cost. To address the objective discrepancy between the reconstruction-oriented training of VAE and the specific requirements of image fusion, we propose a two-stage training strategy to enhance the VAE's adaptability to fusion tasks.

- Extensive experiments demonstrate that our method significantly improves inference speed compared to other diffusion-based approaches. Meanwhile, it also achieves superior fusion performance and shows strong adaptability across various fusion tasks, demonstrating excellent generalization capability..

## 2 Related works

In this section, we first review influential image fusion algorithms from recent years. Then, we introduce the applications of Rectified Flow in various fields, especially in low-level vision.

**Image fusion** Image fusion combines images from different modalities to create a single image with complementary information. Traditional methods [19, 20, 21] use convolutional neural networks to achieve image fusion. The Transformer [22, 23], has also advanced the field of image fusion, particularly when combined with CNNs for multi-modal fusion [24]. Recently, diffusion models have gained attention in low-level vision tasks [25, 26, 15, 16] for their strong generative power, also being applied to image fusion tasks. DDFM [15], using a Denoising Diffusion Probabilistic Model, has shown promising results in infrared-visible and medical image fusion but struggles with adapting to different scenarios. To address this, CCF [16] proposed a controllable diffusion-based fusion framework, which can optimize the fusion process but still faces challenges like excessive sampling steps.

**Rectified flow** Liu [27] first proposed the Rectified Flow method, which generates high-quality images by straightening the path between two data distributions, requiring only one or a few sampling steps. InstaFlow [28] applies Rectified Flow to text-to-image (T2I) models, using the same approach to straighten the trajectories of probability flows, enabling it to generate high-quality images in a single step. FlowGrad [29] backpropagates gradients along the ODE trajectory, effectively enabling control over the generated content of a pre-trained Rectified Flow model. Recently, some Rectified Flow-based methods [30, 31, 32, 33] have also been applied to low-level vision tasks for model acceleration. FlowIE [31] constructs a linear many-to-one transport mapping using conditioned Rectified Flow to achieve efficient image enhancement. FluxSR [30] leverages Rectified Flow to distill diffusion model priors, enabling one-step real-world image super-resolution.

## 3 Preliminary

**Rectified Flow** Traditional diffusion models are trained by predicting the noise added during the forward process, enabling the model to generate high-quality images from Gaussian noise. However, this process typically requires multiple sampling steps, which significantly prolongs the inference time. In general, the forward process can be represented as:

$$x_t = a_t x_0 + b_t \epsilon, \quad \epsilon \sim \mathcal{N}(0, 1). \tag{1}$$

$a_t$ and $b_t$ satisfy $a_t = 1$, $b_t = 0$ and $a_t = 0$, $b_t = 1$. In DDPM, this formula can be expressed as:

$$x_t = \sqrt{\bar{a}_t} x_0 + \sqrt{1 - \bar{a}_t} \epsilon, \quad \epsilon \sim \mathcal{N}(0, 1). \tag{2}$$

Different from DDPM, Rectified Flow treats the forward process as a transformation between two data distributions, which can be seen as a transformation between Gaussian noise and the real image distribution in this case. The goal of Rectified Flow is to train a model $v_\theta$ to predict the velocity $v_t(x_t)$ along the path at step $t$ as

$$\mathcal{L}_{\text{RF}}(\theta) = \mathbb{E}_{t,x_t} \| v_\theta(x_t, t) - v_t(x_t) \|^2. \tag{3}$$

Rectified Flow views the forward process as a straight path between the real data distribution and the noise distribution, and its noise addition formula can be derived through linear interpolation. According to Equation 1, we can derive:

$$x_t = (1 - t)x_0 + t\epsilon, \quad \epsilon \sim \mathcal{N}(0, 1). \tag{4}$$

In this case, $v_t(x_t)$ can be expressed as:

$$x'_t = v_t(x_t) = \frac{\epsilon - x_t}{1 - t} = \epsilon - x_0. \tag{5}$$

Therefore, according to Equation 3, the training objective of Rectified Flow can be derived as:

$$\mathcal{L}_{\text{RF}}(\theta) = \mathbb{E}_{t,x_t,\epsilon} \| v_\theta(x_t, t) - (\epsilon - x_0) \|_2^2. \tag{6}$$

By training a neural network on a large-scale dataset, the output of the network, $v_\theta(x_t, t)$, is encouraged to closely match the training target $\epsilon - x_0$. This enables the model to find the shortest path between two data distributions, significantly accelerating the sampling process.

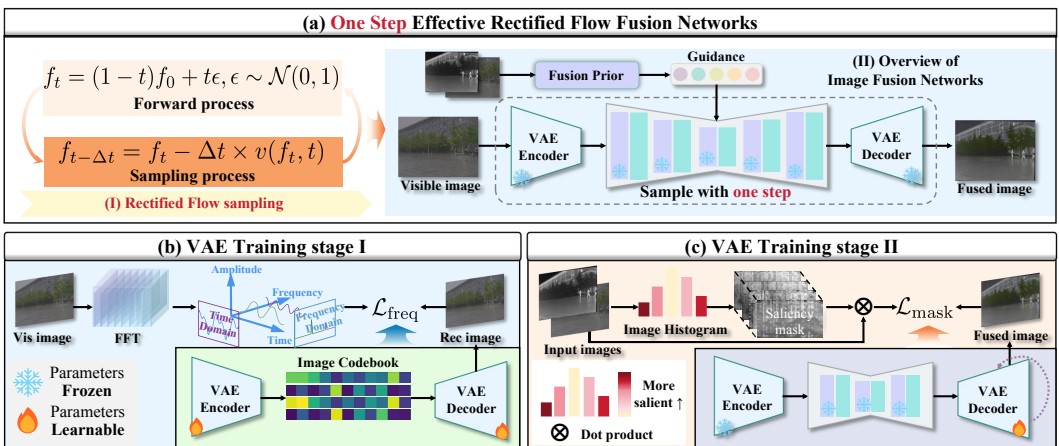

Figure 2: Illustration of the train and inference pipeline of our methods.

**Variational Autoencoder** In diffusion models, Variational Autoencoders (VAEs) are commonly employed to learn low-dimensional latent representations of data. By encoding images into latent spaces, VAEs enable the diffusion process to operate in a more compact latent space, significantly reducing computational costs while improving modeling efficiency. The training objective of a VAE can be formulated as:

$$\mathcal{Q}^* = \arg \min_{E,G,\mathcal{Z}} \max_{D} \mathbb{E}_{x \sim p(x)} \left[ \mathcal{L}_{\mathrm{VQ}}(E, G, \mathcal{Z}) + \lambda \mathcal{L}_{\mathrm{GAN}}(\{E, G, \mathcal{Z}\}, D) \right], \tag{7}$$

where $E$ and $G$ represent the encoder and decoder, respectively, $\mathcal{Z}$ denotes the discrete codebook, and $D$ is the GAN discriminator. This formulation allows the VAE to produce compact yet semantically meaningful latent codes, which serve as an efficient and expressive latent space for subsequent diffusion modeling.

## 4 Methods

In this section, we first introduce how existing methods utilize diffusion models to achieve image fusion. Then, we describe how we incorporate Rectified Flow into the fusion task. Finally, we present the task-specific Variational Autoencoder (VAE) specifically designed for the fusion task, including the two-stage training strategy of VAE and the loss function used for guidance. The pipeline of our method is illustrated in Figure 2.

### 4.1 Implementation of Fusion Methods in Diffusion Models

Previous image fusion methods based on diffusion models typically leverage prior knowledge acquired through pre-trained diffusion models to generate high-quality fused images. Inspired by the work presented in [34], these methods incorporate fusion image information into the sampling process via posterior sampling mechanisms of diffusion models, thus effectively guiding the generation of the fused image. During this sampling process, fusion information is progressively integrated and validated, ultimately achieving high-quality image fusion. The specific mathematical formulation can be expressed as follows:

$$p_\theta \left( f_{(0:T)} \mid i, v \right) = p(f_T) \prod_{t=1}^{T} p_\theta \left( f_{t-1} \mid f_t, i, v \right), \tag{8}$$

where $f_0$ is the fused result and $f_T$ is the initial sampling image, usually Gaussian noise. Additionally, the corresponding posterior sampling can be solved using a Stochastic Differential Equation (SDE), and through Bayes' theorem, we can derive:

$$\nabla_{f_t} \log p_t(f_t \mid i, v) = \nabla_{f_t} \log p_t(f_t) + \nabla_{f_t} \log p_t(i, v \mid f_t), \tag{9}$$

where $\nabla_{f_t} \log p_t(f_t)$ can be obtained via the SDE formulation, inspired by [34], $\nabla_{f_t} \log p_t(i, v \mid f_t)$ can be expressed as:

$$\nabla_{f_t} \log p_t(i, v \mid f_t) \approx \nabla_{f_t} \log p_t(i, v \mid \tilde{f}_{0|t}) \approx \rho \nabla_{f_t} \| i, v - \mathcal{M}(\hat{f}_0(f_t)) \|_2^2. \tag{10}$$

Therefore, the fusion prior can be incorporated into the sampling process by computing the observations $\|i, v - \mathcal{M}(\hat{f}_0(f_t))\|$ between the fused image and the input images. In this manner, the high-quality image generation capability of diffusion models is effectively leveraged to achieve image fusion. A more detailed derivation of the formulas is provided in Appendix A.

## 4.2  One Step Effective Fusion Network

To enable efficient one-step image fusion, inspired by [35], we adopt Rectified Flow for the fusion task. Specifically, we utilize a pre-trained model based on Rectified Flow sampling to generate high-quality fused images. Notably, we observe that using visible-light images as input, rather than pure Gaussian noise, leads to improved fusion performance. The sampling process in our method can be formally expressed as:

$$f_{t-\Delta t} = f_t - \Delta t \times v(f_t, t) \quad v_t(f_t) = \frac{\epsilon - f_t}{1 - t} \quad t \in [1, 0]. \tag{11}$$

Followed by DDFM [15], we incorporate the inference results of the Expectation-Maximization (EM) algorithm into the sampling process of the diffusion model, thereby injecting the prior of the fused image into the diffusion model through posterior sampling, and achieving image fusion. This process can be formulated as:

$$p_\theta(f_0 \mid i, v) = \int p(f_t) \, \delta \left( f_0 - (f_t - \Delta t \cdot v_\theta(f_t \mid i, v)) \right) df_t, \tag{12}$$

where $p(f_t)$ denotes the initial distribution, while the Dirac delta function $\delta$ ensures that the output $f_0$ is strictly determined by the input $f_t$ and the velocity field $v_\theta(f_t \mid i, v)$. It is important to note that Rectified Flow leverages an Ordinary Differential Equation (ODE) framework, meaning that no stochastic noise is injected during the sampling process. Instead, data is deterministically transformed from an initial distribution to the target distribution by optimizing a continuous velocity field $v_\theta(f_t|i, v)$. Consequently, Equation 9 in our method can be reformulated as

$$v_\theta(f_t|i, v) = v_\theta(f_t) + \nabla_{f_t} \log p(i, v \mid f_t) \approx v_\theta(f_t) + \nabla_{f_t} \log p_t(i, v \mid \tilde{f}_{0|t}). \tag{13}$$

In this way, we transfer Rectified Flow to the image fusion task, achieving an efficient single-step image fusion method without requiring additional training.

## 4.3  VAE Autoencoder for Image Fusion

LDM [26] was the first to introduce generative diffusion models into the latent space, leveraging the powerful encoding capability of Variational Autoencoder (VAE) to perform image tasks in the latent space, significantly reducing inference costs while achieving high visual fidelity.

Inspired by this work, we introduce VAE into the image fusion task to enable image generation in the latent space. However, applying VAE-based approaches to image fusion faces two key challenges: (1) Previous methods for image reconstruction typically focus on pixel-level visual fidelity, whereas the core of image fusion lies in capturing complementary semantic information across different modalities. (2) Unlike traditional reconstruction tasks where the objective is to recover the original input, image fusion requires decoding a fused image that integrates information from multiple input modalities. Due to inherent differences between the input images and the desired fused output, this discrepancy poses a significant challenge for the direct application of pretrained VAE in image fusion tasks. To address the aforementioned challenges, we propose a two-stage training strategy to effectively adapt VAE architectures for the image fusion task.

**VAE training stage I** To address Challenge I, we devise a training strategy based on frequency similarity. Specifically, prior research has demonstrated that the complementary semantic information emphasized in image fusion is often closely correlated with the high- and low-frequency components of the input images. Leveraging this insight, we introduce a frequency similarity loss and fine-tune only the VAE encoder and decoder, without involving Rectified Flow sampling or the image fusion process. As a result, the training procedure closely resembles that of conventional image reconstruction. Followed by Equation 7, the corresponding training goal is formulated as follows:

$$\mathcal{R} = \arg \min_{E, G, \mathcal{Z}, x} \max_{D} \mathbb{E}_{x \sim p(x)} \left[ \mathcal{L}_{\text{VQ}}(E, G, \mathcal{Z}) + \lambda_{\text{GAN}} \mathcal{L}_{\text{GAN}}(\{E, G, \mathcal{Z}\}, D) + \lambda_{\text{fre}} \mathcal{L}_{\text{fre}}(x, \hat{x}) \right], \tag{14}$$

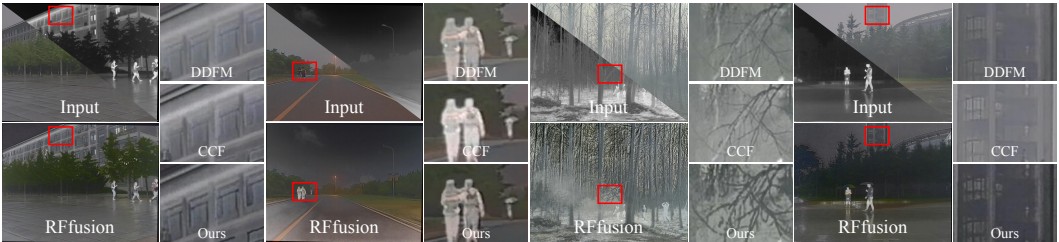

Figure 3: Visual comparison of IVIF with SOTA methods on M³FD datasets.

where $\mathcal{L}_{\text{VQ}}$, $\mathcal{L}_{\text{GAN}}$ are followed by [36]. The proposed frequency loss, $\mathcal{L}_{\text{fre}}$, is designed to capture discrepancies in the frequency domain by first transforming the input images from the spatial domain using the Fast Fourier Transform (FFT). The transformation process can be expressed as:

$$\hat{I}_{\text{in}} = \mathcal{F}(I_{\text{in}}), \hat{I}_{\text{rec}} = \mathcal{F}(I_{\text{rec}}), \mathcal{F}(u,v) = \sum_{x=0}^{H-1}\sum_{y=0}^{W-1} I(x,y) \cdot e^{-i\frac{2\pi ux}{H}} \cdot e^{-i\frac{2\pi vy}{W}}. \tag{15}$$

Next, we shift the zero-frequency components of both $\hat{I}_{\text{in}}$ and $\hat{I}_{\text{rec}}$ which represent the average intensity of the images to the center of the spectrum, resulting in $\hat{I}_{\text{in}}$ and $\hat{I}_{\text{rec}}$ for loss computation. The loss can then be formulated as:

$$\mathcal{L}_{\text{fre}} = \left( N\left(\log(1+|\hat{I}_{\text{in}}^{\text{shift}}|)\right) - N\left(\log(1+|\hat{I}_{\text{rec}}^{\text{shift}}|)\right)\right)^2, \tag{16}$$

where $N(\cdot)$ denotes a normalization operation. Optimizing the above losses encourages the VAE to focus on semantic information relevant to fusion during image reconstruction.

**VAE training stage II** To address Challenge II, we propose a training strategy for Variational Autoencoder (VAE) tailored to the image fusion task. Specifically, we integrate the VAE into the overall fusion framework and perform joint training to enhance its adaptability to the fusion process. It is important to note that, in our method, the input is a visible image, and prior information from the fused image is incorporated during the sampling stage to achieve the fusion. As a result, the VAE encoder is only required to effectively compress the input image, while the decoder is responsible for both reconstructing the image and incorporating fusion-related information. Therefore, in the second stage of training, we focus on fine-tuning the VAE decoder to improve its ability to reconstruct fused images. During this stage, we employ a fusion-specific loss function commonly used in image fusion tasks to optimize the VAE, which is formulated as follows:

$$\mathcal{L}_{\text{fusion}} = \lambda_{\text{int}}\mathcal{L}_{\text{int}} + \lambda_{\text{SSIM}}\mathcal{L}_{\text{SSIM}} + \lambda_{\text{grad}}\mathcal{L}_{\text{grad}} + \lambda_{\text{color}}\mathcal{L}_{\text{color}} + \lambda_{\text{mask}}\mathcal{L}_{\text{mask}}, \tag{17}$$

where $\mathcal{L}_{\text{int}}$, $\mathcal{L}_{\text{SSIM}}$, $\mathcal{L}_{\text{grad}}$, and $\mathcal{L}_{\text{color}}$ are followed by [37]. Meanwhile, to achieve saliency-guided regional fusion, we introduce a saliency mask loss, denoted as $\mathcal{L}_{\text{mask}}$, which can be formulated as:

$$\mathcal{L}_{\text{mask}} = \|\mathcal{W}_v \cdot I_v + \mathcal{W}_{ir} \cdot I_{ir} - I_f\|_1. \tag{18}$$

Among them, $I_{ir}$, $I_v$ represent the input images, and $I_f$ are the fused image. $\mathcal{W}_v$ and $\mathcal{W}_{ir}$ denote the saliency-based weight maps computed from the corresponding input images. The saliency mask loss $\mathcal{L}_{\text{mask}}$ guides the network to focus on salient regions during the fusion process, thereby improving the preservation of complementary information within the fused image. By jointly optimizing this loss with other fusion objectives, the reconstruction capability of the VAE decoder is significantly enhanced, ultimately leading to high-quality image fusion results. More details about the training loss $\mathcal{L}_{\text{fusion}}$ and $\mathcal{L}_{\text{mask}}$ can be found in Appendix B.

## 5 Experiments

**Experiment datasets** We conduct experiments on three representative image fusion tasks: infrared and visible image fusion (IVIF), multi-exposure image fusion (MEF), and multi-focus image fusion (MFF). For the infrared and visible image fusion task, evaluations are performed on three widely-used benchmark datasets: M³FD [1], TNO [38], and RoadScene [39]. For the multi-exposure and multi-focus fusion tasks, we utilize the MEFB [40] and MFIF [10] datasets, respectively. The MFIF dataset includes the Lytro [41], MFFW [42], and MFI-WHU [43] datasets.

Table 1: Comparison of Metrics with Our Baseline Method DDFM [15].

| Methods | M$^3$FD Dataset | | | | | T&R Datasets | | | | |
|---|---|---|---|---|---|---|---|---|---|---|
| | EN | MI | SF | VIF | SSIM | EN | MI | SF | VIF | SSIM |
| DDFM | 6.720 | 2.871 | 9.102 | 0.677 | 0.867 | 7.077 | 1.798 | 8.910 | 0.277 | 0.207 |
| Ours | 6.722 | 3.320 | 9.780 | 0.748 | 0.914 | 7.139 | 2.948 | 12.55 | 0.675 | 0.921 |
| | (+0.002) | (+0.449) | (+0.678) | (+0.071) | (+0.047) | (+0.062) | (+1.150) | (+3.640) | (+0.398) | (+0.714) |

**Implementation Details** The two-stage training of the VAE was conducted entirely on an NVIDIA V100 GPU. In the first stage, the model was trained on the LLVIP [44] and MSRS [45] datasets for 20 epochs. Interestingly, the best validation performance was typically achieved within just 4 to 5 epochs. The second stage involved training exclusively on the MSRS [45] dataset for 40 epochs. The remaining hyperparameters for both stages were configured in accordance with the experimental settings detailed in [26] and [37]. We evaluate our method on all three fusion tasks using the same set of checkpoints, without any task-specific fine-tuning, thereby demonstrating the strong generalization capability of our approach across diverse tasks.

Table 2: Quantitative comparison on M$^3$FD, TNO, and RoadScene datasets. The best and second best results are highlighted in **bold** and underline.

| Dataset | M$^3$FD Dataset | | | | T&R Dataset | | | |
|---|---|---|---|---|---|---|---|---|
| Method | MI ↑ | VIF ↑ | SCD ↑ | EN ↑ | MI ↑ | VIF ↑ | SCD ↑ | EN ↑ |
| U2Fusion [3] | 2.760 | 0.633 | 1.569 | 6.659 | 2.599 | 0.556 | 1.338 | 6.821 |
| YDTR [46] | 3.183 | 0.635 | 1.506 | 6.547 | 2.976 | 0.588 | 1.420 | 6.842 |
| UMFusion [47] | 3.089 | 0.613 | 1.570 | 6.669 | 2.888 | 0.610 | 1.475 | 6.967 |
| ReCoNet [48] | 3.066 | 0.577 | 1.483 | 6.679 | 2.985 | 0.540 | 1.510 | 7.051 |
| LRRNet [49] | 2.805 | 0.566 | 1.463 | 6.437 | 2.766 | 0.508 | 1.558 | 7.118 |
| CoCoNet [50] | 2.631 | 0.729 | 1.772 | 7.738 | 2.579 | 0.568 | 1.782 | 7.735 |
| DDFM [15] | 2.871 | 0.677 | 1.683 | 6.720 | 1.798 | 0.277 | 1.160 | 7.077 |
| **Ours** | **3.320** | **0.748** | 1.574 | 6.722 | 2.948 | **0.675** | 1.639 | 7.139 |

Table 3: Efficiency comparisons with other diffusion-based methods. The best and second best results are highlighted in **bold** and underline.

| Metrics | Methods | | | | | | |
|---|---|---|---|---|---|---|---|
| | DRMF | Dif-Fusion | Diff-IF | Text-DiFuse | DDFM | CCF | Ours |
| SF↑ | 12.57 | 10.42 | 13.90 | 9.319 | 9.689 | 10.14 | **14.00** |
| AG↑ | 4.201 | 4.307 | 5.179 | 3.559 | 3.981 | 3.882 | **5.218** |
| Runtime (s) ↓ | 3.221 | 1.997 | 2.457 | 9.199 | 22.03 | 62.47 | **0.308** |
| Parameters (M) | 170.98 | 416.47 | 23.47 | 119.49 | 552.81 | 552.81 | 65.57 |

## 5.1 Experiments on Infrared and Visible Image Fusion

In this section, we conduct a comprehensive comparison between our proposed RFfusion method and other fusion approaches. We begin by comparing it with our baseline method, DDFM [15]. Subsequently, we evaluate its performance against several state-of-the-art methods proposed in recent years to demonstrate the superiority of our approach, including: U2Fusion [3], YDTR [46], UMFusion [47], ReCoNet [48], LRRNet [49], CoCoNet [50], and DDFM [15].

**Comparison with DDFM method** Since our method is built upon the DDFM framework by introducing fusion priors to achieve image fusion—with the main differences lying in the sampling strategy and the use of VAE for latent space generation—we primarily compare our approach with DDFM. As shown in Table 1, our method significantly accelerates inference and reduces computational overhead, while outperforming DDFM across all fusion metrics on multiple datasets. These results indicate that our method not only effectively reduces the number of sampling steps but also substantially enhances the quality of the fused images. Furthermore, the results validate the generality and flexibility of our approach, demonstrating its potential to serve as a plug-and-play module that can be integrated into

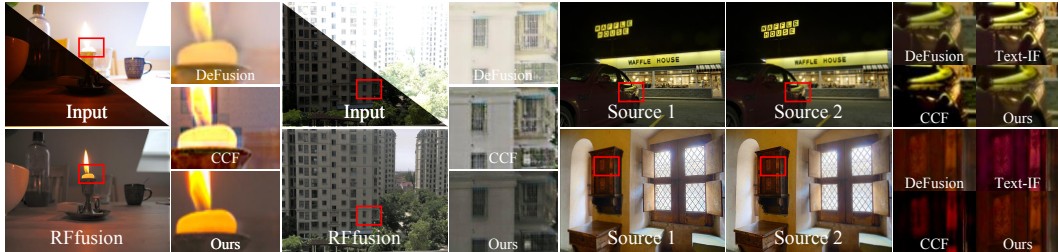

Figure 4: Visual comparison of multi-exposure image fusion and multi-focus image fusion with SOTA methods on MEFB and MFIF datasets.

other diffusion-based image fusion frameworks to simultaneously improve both inference efficiency and fusion performance.

**Quantitative Comparison** As shown in Table 2, we conducted a comprehensive evaluation of the proposed method using four widely adopted quantitative metrics across three benchmark datasets: $M^3FD$, TNO, and Roadscene. The results demonstrate that our method consistently ranks among the top two across most metrics, indicating strong overall performance. Specifically, on the $M^3FD$ dataset, our method achieves the best performance in terms of Mutual Information (MI), highlighting its effectiveness in preserving informative content from the source images. Moreover, our method achieves the highest scores in Visual Information Fidelity (VIF) across all datasets, further confirming its superiority in enhancing the visual quality of the fused images.

**Qualitative Comparison** As shown in Figure 3, our method demonstrates superior visual performance compared to other approaches. We selected four images from the $M^3FD$ dataset for qualitative analysis, covering different scenarios including both daytime and nighttime, to ensure a comprehensive evaluation. Our method better preserves detailed texture information from the original images, such as window textures on buildings and fine details at the ends of tree branches, whereas other methods tend to blur these features. Additionally, our method highlights human details more effectively and better retains the mutual information between visible and infrared images. This demonstrates the advantages of our method in qualitative results.

## 5.2 Experiments on efficiency comparisons with other diffusion-based methods

To verify the effectiveness of our method in reducing the inference time and computational cost of diffusion models in image fusion tasks, we compare it with several diffusion-based image fusion approaches proposed in recent years, including DRMF [51], Dif-Fusion [18], Diff-IF [52], Text-DiFuse [17], DDFM [15], and CCF [16]. All experiments are conducted on an NVIDIA V100 GPU, and the fusion speed as well as the number of model parameters are evaluated on the RoadScene [39] dataset to comprehensively assess the efficiency and complexity of each method. As shown in Table 3 and Figure 1, compared with other diffusion-based image fusion methods, our approach demonstrates a significant advantage in inference speed while also achieving superior fusion quality. These results indicate that our method not only greatly improves inference efficiency but also maintains, or even enhances fusion performance, fully validating its capability for joint optimization of efficiency and effectiveness.

## 5.3 Experiments on Evaluation on Multi-Focus Fusion

**Quantitative Comparison** As shown in Table 6, we conducted a comprehensive evaluation of the proposed method on the MFIF dataset. The experimental results demonstrate that our method consistently ranks among the top two across most evaluation metrics, fully validating its superior effectiveness in the multi-focus fusion (MFF) task. Notably, the proposed method achieves this performance without any fine-tuning on the multi-focus fusion dataset, further confirming its strong generalization ability and robustness across different tasks.

**Qualitative Comparison** As shown in Figure 6, we selected two representative images from the MFIF dataset for qualitative analysis, covering both daytime and nighttime scenarios to ensure a comprehensive evaluation. Compared to other methods, our method more effectively preserves

Table 4: Quantitative comparison on MEFB and MFIF datasets. The best and second best results are highlighted in **bold** and underline.

| Dataset | MEFB Dataset | | | | MFIF Dataset | | | |
|---|---|---|---|---|---|---|---|---|
| Method | MI ↑ | CC ↑ | Qcb ↑ | PSNR ↑ | MI ↑ | CC ↑ | Qcb ↑ | PSNR ↑ |
| DeFusion [53] | 4.854 | 0.834 | 0.365 | 57.70 | 6.007 | 0.976 | 0.627 | **76.62** |
| TC-MoA [54] | 5.418 | 0.900 | 0.430 | **59.00** | **6.686** | 0.968 | **0.731** | 74.78 |
| Text-IF [37] | 5.596 | 0.860 | 0.385 | 56.44 | 5.399 | 0.967 | 0.629 | 71.74 |
| DDFM [15] | 3.850 | 0.792 | 0.321 | 58.39 | 3.232 | 0.772 | 0.413 | 66.24 |
| CCF [16] | 4.830 | 0.898 | 0.398 | 58.38 | 4.799 | 0.956 | 0.474 | 66.64 |
| **Ours** | **6.528** | **0.901** | **0.461** | 58.49 | 6.443 | **0.977** | 0.654 | 75.04 |

clear details from the original images, achieving high-quality multi-focus image fusion and fully demonstrating its superior performance in qualitative evaluation.

### 5.4 Experiments on Evaluation on Multi-Exposure Fusion

**Quantitative Comparison** Table 6 presents a quantitative comparison between our method and existing approaches on the MEFB dataset. Notably, our method does not require any fine-tuning on the MEF dataset. It consistently outperforms other multi-exposure fusion (MEF) methods across most evaluation metrics, demonstrating its superior performance and strong generalization capability in the multi-exposure image fusion task.

**Qualitative Comparison** As shown in Figure 4, compared to other methods, our fusion results better preserve the detailed features of the original images and achieve superior visual performance. We selected two images from the MEFB dataset for qualitative analysis. Our method more effectively retains the texture of windows and the contour features of candles, demonstrating its advantages in qualitative evaluation.

Table 5: Ablation studies on the effectiveness of the two-stage training strategy and loss functions.

| Stage I | Stage II | PSNR | MI | SF | AG | | $\mathcal{L}_{\text{fre}}$ | $\mathcal{L}_{\text{mask}}$ | PSNR | MI | SF | AG |
|---|---|---|---|---|---|---|---|---|---|---|---|---|
| – | – | 59.41 | 2.998 | 12.16 | 4.615 | | – | – | 57.22 | 2.944 | 12.77 | 4.882 |
| ✓ | – | 59.68 | 3.017 | 12.88 | 4.676 | | ✓ | – | 58.84 | 3.202 | 13.56 | 5.021 |
| – | ✓ | 60.36 | 3.001 | 12.57 | 4.783 | | – | ✓ | 59.67 | 3.121 | 13.31 | 4.976 |
| ✓ | ✓ | **61.81** | **3.220** | **14.00** | **5.218** | | ✓ | ✓ | **61.81** | **3.220** | **14.00** | **5.218** |

### 5.5 Ablation Study

**Experimental on the effectiveness of the two-stage training strategy.** We conducted ablation studies on the two-stage training strategy for the VAE to evaluate its effectiveness in the image fusion task. As shown in Table 5, we compared fusion performance under four settings: without any training, using only the first-stage training, using only the second-stage training, and applying both stages of training. The results demonstrate that the fusion performance is optimal when both stages are applied, validating the effectiveness of the training strategy in enhancing fusion quality.

**Experimental on the effectiveness of the loss functions.** We conducted ablation experiments on the loss functions used in our proposed training method to evaluate their contributions to image fusion performance, as shown in Table 5. Specifically, we designed four experimental settings: without using either $\mathcal{L}_{\text{fre}}$ or $\mathcal{L}_{\text{mask}}$, using only $\mathcal{L}_{\text{fre}}$, using only $\mathcal{L}_{\text{mask}}$, and using both $\mathcal{L}_{\text{fre}}$ and $\mathcal{L}_{\text{mask}}$ simultaneously. The experimental results demonstrate that both $\mathcal{L}_{\text{fre}}$ and $\mathcal{L}_{\text{mask}}$ can independently improve fusion quality, while their combined use leads to the best performance. These findings validate the effectiveness and necessity of the proposed loss function design.

## 6 Limitation

RFfusion still relies on a Rectified Flow pre-trained model trained on generic image generation tasks, which is not specifically designed for image fusion. This limitation may hinder the further improvement of RFfusion's fusion performance.

# 7 Conclusion

In this paper, we propose an efficient one-step diffusion-based image fusion method called RFfusion. By integrating Rectified Flow into the image fusion task, our method leverages its efficient one-step sampling mechanism to significantly accelerate the diffusion-based fusion process. Moreover, we design a task-specific Variational Autoencoder (VAE) that performs fusion in the latent space, effectively reducing computational overhead while preserving more image details. Extensive experimental results demonstrate that RFfusion achieves superior performance in both inference speed and fusion quality compared to existing state-of-the-art methods, and also exhibits strong generalization capabilities across diverse image fusion tasks. In the future, we will further explore acceleration mechanisms of diffusion models in image fusion tasks to achieve more efficient image fusion methods

**Acknowledgments:** This work was partially supported by the National Natural Science Foundation of China (No.62302078 and No.62372080), and China Postdoctoral Science Foundation (2023M730741).

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

## Appendix

## A    Detailed derivation of formulas

In this section, we provide a detailed proof regarding posterior sampling in diffusion-based image fusion methods. The noise predicted by the diffusion model at time step $t$ is often related to the score at the current time step. According to [55], the specific formulation can be expressed as:

$$\epsilon_\phi(x_t, t) = -\sqrt{1-\alpha_t}\nabla_{x_t}\log p(x_t), \tag{19}$$

In posterior sampling, we also need to take into account the guidance from image fusion, denoted as $i, v$. Therefore, what we need to solve is $\nabla_{f_t}\log p(f_t|i,v)$, which can be expressed using Bayes' theorem as:

$$p_t(f_t \mid i, v) = \frac{p_t(i, v \mid f_t) \cdot p_t(f_t)}{p_t(i, v)},$$
$$\Rightarrow \log p_t(f_t \mid i, v) = \log p_t(i, v \mid f_t) + \log p_t(f_t) - \log p_t(i, v),$$
$$\Rightarrow \nabla_{f_t}\log p_t(f_t \mid i, v) = \nabla_{f_t}\log p_t(f_t) + \nabla_{f_t}\log p_t(i, v \mid f_t). \tag{20}$$

Among them, $\nabla_{f_t}\log p(f_t \mid i, v)$ and $\nabla_{f_t}\log p(f_t)$ can be expressed by Equation 19 as follows:

$$\epsilon_\phi(f_t, t) = -\sqrt{1-\alpha_t}\nabla_{f_t}\log p_t(f_t),$$
$$\epsilon'_\phi(f_t, t \mid i, v) = -\sqrt{1-\alpha_t}\nabla_{f_t}\log p_t(f_t \mid i, v). \tag{21}$$

Therefore, the final equation can be expressed as:

$$\epsilon'_\phi = \epsilon_\phi(f_t, t) - \sqrt{1-\alpha_t}\nabla_{f_t}\log p_t(i, v \mid f_t),$$
$$\approx \epsilon_\phi(f_t, t) - \sqrt{1-\alpha_t}\nabla_{f_t}\log p_t(i, v \mid \tilde{f}_{0|t}),$$
$$\approx \epsilon_\phi(f_t, t) - \rho\sqrt{1-\alpha_t}\nabla_{f_t}\|i, v - \mathcal{M}(\hat{f}_0(f_t))\|_2^2. \tag{22}$$

Therefore, we inject the image fusion prior by correcting the predicted noise during the sampling process, thereby achieving high-quality image fusion based on the diffusion model.

## B    Details about the training loss

In the main text, the loss $\mathcal{L}_{\text{fusion}}$ used in the training stage II is defined as follows:

$$\mathcal{L}_{\text{fusion}} = \lambda_{\text{int}}\mathcal{L}_{\text{int}} + \lambda_{\text{SSIM}}\mathcal{L}_{\text{SSIM}} + \lambda_{\text{grad}}\mathcal{L}_{\text{grad}} + \lambda_{\text{color}}\mathcal{L}_{\text{color}} + \lambda_{\text{mask}}\mathcal{L}_{\text{mask}}. \tag{23}$$

Followed by [37], we use the Intensity Loss to encourage the model to focus on salient features in the fused image, which is specifically defined as:

$$\mathcal{L}_{\text{int}} = \frac{1}{HW}\left\| I_f - \max(I_{\text{vis}}^g, I_{\text{ir}}^g) \right\|_1. \tag{24}$$

Here, $I_{\text{vis}}^g$ and $I_{\text{ir}}^g$ are the ground truth corresponding to the fused image. We also use $\mathcal{L}_{\text{SSIM}}$ to train the model so that the fused image is structurally as similar as possible to the two input images. It is defined as:

$$\mathcal{L}_{\text{SSIM}} = (1 - SSIM(I_f, I_{\text{vis}}^g)) + \mu(1 - SSIM(I_f, I_{\text{ir}}^g)). \tag{25}$$

We also compute the gradient loss $\mathcal{L}_{\text{grad}}$ to ensure the similarity between the fused image and the input images in terms of edge features:

$$\mathcal{L}_{\text{grad}} = \frac{1}{HW}\left\| \nabla I_f - \max(\nabla I_{\text{vis}}^g, \nabla I_{\text{ir}}^g) \right\|_1, \tag{26}$$

and use $\mathcal{L}_{\text{color}}$ to keep consistent color with input images:

$$\mathcal{L}_{\text{color}} = \frac{1}{HW}\left\| \mathcal{F}_{CbCr}(I_f) - \mathcal{F}_{CbCr}(I_{\text{vis}}^g) \right\|_1. \tag{27}$$

As mentioned in the main text, we use $\mathcal{L}_{\text{mask}}$ for saliency-guided regional fusion, which can be represented as:

$$\mathcal{L}_{\text{mask}} = \left\| \mathcal{W}_v \cdot I_v + \mathcal{W}_{ir} \cdot I_{ir} - I_f \right\|_1. \tag{28}$$

Here, $\mathcal{W}_v$ and $\mathcal{W}_{ir}$ denote the saliency-based weight maps computed from the corresponding input images. $\mathcal{W}_v$ and $\mathcal{W}_{ir}$ are computed based on pixel-level visual saliency maps. Specifically, they are estimated by measuring the sparsity of the image pixel histograms: the sparser the pixel distribution, the higher the corresponding saliency. The computation can be formulated as:

$$\text{Saliency}(i) = \sum_{j=0}^{255} |i - j| \cdot \text{Hist}(j).$$  (29)

Here, $i$ represents the grayscale value of the current pixel, and $j$ denotes the grayscale value of the traversed pixels. By computing the saliency values $S_{ir}(i,j)$ and $S_v(i,j)$ for each pixel, we can obtain the corresponding $\mathcal{W}_v$ and $\mathcal{W}_{ir}$. The specific formulas are given as:

$$\mathcal{W}_v(i,j) = \mu_v + \mathcal{S}_v(i,j) - \mu_v \cdot \mathcal{S}_{ir}(i,j)$$  (30)

$$\mathcal{W}_{ir}(i,j) = 1 - \mathcal{W}_v(i,j)$$  (31)

## C  Additional experiments of our method

Table 6: Quantitative comparison on Lytro, MFFW and MFI-WHU datasets. The best and second best results are highlighted in **bold** and underline.

| Dataset | Lytro | | | | MFFW | | | | MFI-WHU | | | |
|---|---|---|---|---|---|---|---|---|---|---|---|---|
| Method | MI | CC | Qcb | PSNR | MI | CC | Qcb | PSNR | MI | CC | Qcb | PSNR |
| DeFusion [53] | 6.27 | 0.97 | 0.59 | **77.2** | 5.59 | 0.97 | 0.55 | **74.4** | 6.01 | 0.97 | 0.69 | **77.2** |
| TC-MoA [54] | **7.45** | 0.97 | **0.76** | 74.8 | 5.34 | 0.96 | **0.63** | 72.8 | **6.76** | 0.97 | **0.75** | 75.6 |
| Text-IF [37] | 5.63 | 0.97 | 0.65 | 71.9 | 5.26 | 0.96 | 0.61 | 70.2 | 5.31 | 0.97 | 0.62 | 72.3 |
| DDFM [15] | 3.53 | 0.85 | 0.41 | 67.2 | 3.33 | 0.73 | 0.38 | 64.5 | 2.99 | 0.74 | 0.43 | 66.4 |
| CCF [16] | 5.15 | 0.96 | 0.49 | 66.8 | 4.47 | 0.95 | 0.47 | 66.6 | 4.95 | 0.96 | 0.47 | 67.1 |
| **Ours** | 6.58 | **0.98** | 0.61 | 76.1 | **5.80** | **0.97** | 0.55 | 71.7 | 6.38 | **0.98** | 0.71 | 75.7 |

**More Quantitative Comparison of Multi-Focus Fusion** As shown in Table 6, we conducted comparisons on three datasets in MFIF: Lytro [41], MFFW [42], and MFI-WHU [43]. Our method outperforms other approaches on most metrics. Specifically, it achieves either the first or second best performance[2] on 9 different metrics, surpassing other comparison methods and demonstrating the superiority of our approach in the Multi-Focus Fusion task.

---

[2]Compare the percentiles of the same value.

