# OpenReview forum: "Efficient Rectified Flow for Image Fusion"
_NeurIPS.cc/2025/Conference — NeurIPS 2025 poster_

### Official Review · Reviewer_eb1j · 2025-06-30

**Clarity:** 4
**Significance:** 3
**Originality:** 4
**Rating:** 5
**Confidence:** 4

**Summary:**

This paper proposes RFfusion, an efficient one-step image fusion framework that integrates Rectified Flow into multimodal fusion tasks. By combining a task-specific latent-space VAE and a two-stage training strategy, the method achieves fast inference and high-quality results across infrared-visible, multi-exposure, and multi-focus fusion benchmarks.

**Questions:**

Please refer to the weaknesses.

**Ethical Concerns:**

["NO or VERY MINOR ethics concerns only"]

**Final Justification:**

The authors have sufficiently responded to my concerns. I have therefore raised my score to Accept.

**Limitations:**

Yes.

**Quality:**

3

**Strengths And Weaknesses:**

Strengths:
1.Integrating Rectified Flow into multimodal image fusion is interesting and technically reasonable.
2.The method shows consistently strong performance across multiple fusion tasks.
3.The paper is well-organized, with clear methodology, and solid experiments.
4.The proposed two-stage latent space training strategy is practically appealing.

Weaknesses:
1.The method does not incorporate any explicit mechanisms to model modality-specific characteristics, such as dual-branch encoders or cross-modality attention. Would this design limit the model’s ability to preserve complementary features from each modality?
2.The VAE encoder exclusively takes the visible image as input, while the infrared modality is introduced only via posterior gradients during sampling. I am curious whether may result in biased latent representations and insufficient encoding of infrared-specific semantics.
3.Equation (10) introduces a posterior regularization term involving a mapping function, yet the formulation and implementation of M are not clearly specified. It is unclear whether this component is learned, fixed, or task-specific, making the role of the fusion prior difficult to assess.
4.In Equation (13), the fusion velocity is defined as a sum of the pre-trained rectified flow and a posterior gradient term. However, the dynamics of their interaction are not analyzed. It remains unclear whether this additive formulation can consistently preserve modality alignment or may distort flow trajectories under large posterior corrections.

---

> ### Author Rebuttal · Authors · 2025-07-29
>
> Thank you for your valuable comments, especially for recognizing the **technical merits** and **performance of our method**. Below are our responses to your questions.
>
> - **W1: The method does not incorporate any explicit mechanisms to model modality-specific characteristics.**
>
> Thanks for your comment. We acknowledge that, compared to our method, dual-branch encoders or cross-modality attention modules offer a more intuitive advantage for image fusion tasks, as they explicitly integrate features from different modalities. However, we would like to emphasize that input images often suffer from quality issues such as noise and blur. Our method leverages the capability of generative models to enhance image quality during the fusion process—something traditional fusion modules are typically unable to achieve. Furthermore, as shown in **Tables 2** and **4** of our paper, our method achieves superior performance in terms of the **MI** metric, indicating that it effectively preserves critical information from the source images even without relying on conventional fusion modules.
>
> - **W2: Indirect infrared input may bias the latent space and weaken infrared feature encoding.**
>
> Our method achieves leading performance in terms of the **MI** metric across multiple datasets. The **MI** metric typically reflects the richness of information from the input images contained in the fused image, which indicates that our method does not suffer from insufficient encoding of infrared-specific semantics.
>
> - **W3: The unclear definition and learning of $M$.**
>
> Here, $M$ can be either a linear or a nonlinear operation. It encompasses the learned, fixed, and task-specific components mentioned by the reviewer. In this paper, we adopt the **Expectation-Maximization** algorithm used in DDFM [1]. However, we do not intend to limit it to a single algorithm. For example, in CCF [2], multiple types of conditions are employed, such as **Basic Conditions**, **Enhanced Conditions**, and **Task-specific Conditions**.
>
> - **W4: The additive fusion in Equation (13) lacks analysis.**
>
> Our formulation defines the fusion velocity as an additive combination of the pretrained Rectified Flow $v_\theta(f_t)$ and a posterior correction term $\nabla_{f_t} \log p(i,v \mid f_t)$, which serves to guide the generative process toward better alignment with the input modalities. While this additive form may appear simplistic, it is inspired by techniques in **score-based generative modeling**, where posterior guidance is often introduced as a gradient correction to the original flow or score field. Empirically, we found this formulation to be stable during generation, as the pretrained flow already provides a strong data prior, and the posterior term acts as a relatively localized correction based on specific input instances.
>
> **Reference**
>
> [1] Zhao Z, Bai H, Zhu Y, et al. DDFM: denoising diffusion model for multi-modality image fusion[C]//Proceedings of the IEEE/CVF international conference on computer vision. 2023: 8082-8093.
>
> [2] Cao B, Xu X, Zhu P, et al. Conditional controllable image fusion[J]. Advances in Neural Information Processing Systems, 2024, 37: 120311-120335.

---

> > ### Author Response · Authors · 2025-08-03
> >
> > Thank you again for the reviewer's valuable input throughout this review process. To ensure we make the most of the remaining time in the discussion phase, we would greatly appreciate it if you could let us know whether any questions remain or if our previous responses have resolved the issues raised. We are fully committed to addressing any further concerns you may have.

---

> > > ### Comment · Reviewer_eb1j · 2025-08-04
> > >
> > > The authors have sufficiently responded to my concerns. I have therefore raised my score to Accept.

---

### Official Review · Reviewer_oayV · 2025-06-30

**Clarity:** 4
**Significance:** 4
**Originality:** 3
**Rating:** 4
**Confidence:** 4

**Summary:**

This article proposes an efficient one-step image fusion method RFfusion, which combines Rectified Flow and task specific VAE structure, aiming to improve the inference efficiency of image fusion while maintaining high-quality output.

**Questions:**

1. Although experiments have shown that fine-tuning is not necessary in multiple task, could generalization tests (such as MRI-PET mentioned in Section 1 Introduction) be provided under different modal combinations for further validation?

2. Although the effectiveness of the loss term has been demonstrated, could authors further ablate the model architecture (such as different encoder or decoder structures in the proposed VAE module architecture, and the detailed architecture design in the proposed fusion prior guidance)?


3. Authors currently use visible image as the starting point instead of the other modality or the pure noise. Could authors consideration been given to using other starting point (such as dual modal images) to improve fusion consistency?

**Ethical Concerns:**

["NO or VERY MINOR ethics concerns only"]

**Final Justification:**

Thank you for the rebuttal. It effectively addressed my concerns, and I have decided to maintain my accept score.

**Limitations:**

yes

**Quality:**

3

**Strengths And Weaknesses:**

Strength:
1. The RFFusion proposed by the author is the first to apply Rectified Flow to image fusion tasks, proposing a one-step sampling mechanism that significantly reduces inference time.

2. A two-stage VAE training strategy is designed for image fusion, effectively solving the problem of inconsistent reconstruction and fusion targets.

3. Quantitative and qualitative comparisons of various image fusion tasks are conducted (infrared visible light, multi exposure, multi focus) and the method performs excellently.



Weakness:
1. The pre trained Rectified Flow model is not specifically designed for image fusion, which limits further performance improvement.

2. The author did not conduct ablation experiments based on whether Rectified Flow and VAE structure were used, which limited the evaluation of the improvement points.

---

> ### Author Rebuttal · Authors · 2025-07-29
>
> Thank you for your valuable comments and for acknowledging **the effectiveness of our method across various image fusion tasks**. Below are our responses to your questions, and we hope they address your concerns.
>
> - **W1: The pretrained Rectified Flow model is not specifically designed for image fusion.**
>
> Thanks for your comment. We acknowledge that the pretrained Rectified Flow model was not specifically designed for image fusion tasks, and compared to traditional image fusion methods, our approach does not intuitively fuse features from the two source images. However, we would like to emphasize that our method leverages **the powerful generative capability** of Rectified Flow to produce high-fidelity fused images, while incorporating rich information from both images through posterior sampling to ensure the quality of the fusion results. Therefore, the design of our method does not compromise the final performance. Moreover, some approaches utilizing Rectified Flow have also been successfully applied to other low-level vision tasks [1,2,3], which further demonstrates the feasibility of our method.
>
> - **W2: Additional ablation experiments on Rectified Flow and VAE structure.**
>
> We have added ablation studies on VAE and Rectified Flow. For experiments related to inference speed, please refer to weakness 3 mentioned by **reviewer uZpR**. The quantitative results of the ablation studies are shown in the table. We conducted a comparison on the Roadscene dataset.
>
> | VAE | RF | MI | SF | AG | SSIM |
> |--------|--------|--------|--------|--------|--------|
> | - | ✓ | 3.07 | 12.5 | 5.06 | 0.89
> | ✓  | ✓ | **3.22**  | **14.0** | **5.22** | **0.93**
>
> - **Q1: Providing generalization tests under different modal combinations.**
>
> Please refer to weakness 2 mentioned by **reviewer fHBh**.
>
> - **Q2: Further ablation of the model architecture.**
>
> We have added the results of image fusion under different VAE architectures. We conducted experiments using VQ-VAE (with a **downsampling factor of 4** and **without attention**) and trained the VAE structure using the same training procedure.
> The results are shown in the table below.
>
> | M3FD | MI | VIF | SCD | EN | SSIM | PSNR |
> |-------------|--------|--------|--------|--------|--------|--------
> | VAE (no attention)| 3.08 | 0.73  | 1.57  | 6.68 | 0.89  | 62.4  |
> | VAE | **3.32**  | **0.75**  | **1.60**  | **6.72**  | **0.91**  | **63.2**  |
>
> | RoadScene | MI | VIF | SCD | EN | SSIM | PSNR |
> |-------------|--------|--------|--------|--------|--------|--------|
> | VAE (no attention)| 3.02  | 0.64  | 1.55  | 7.19  | 0.91  | 60.4  |
> | VAE |  **3.22**  | **0.67**  | **1.66**  | **7.28**  | **0.93**  | **61.8** |
>
> | TNO | MI | VIF | SCD | EN | SSIM | PSNR |
> |-------------|--------|--------|--------|--------|--------|--------|
> | VAE (no attention)| 2.63  | 0.62  | 1.57  | 6.66  | 0.88  | 61.8 |
> | VAE | **2.68**  | **0.68**  | **1.61** | **6.99**  | **0.91**  | **62.6**  |
>
> As shown in the table, although the **VAE without attention** offers a slight advantage in inference speed, its reconstruction capability is inferior to that of the VAE with attention. Therefore, we choose the **attention-based VAE** as our pretrained model.
>
> - **Q3: Other starting points to improve fusion consistency.**
>
> Regarding why we do not use noise as the starting point, you can refer to Question 1 raised by **Reviewer uZpR**. As for using **dual modal images** as the starting point, since Rectified Flow only accepts a single image as input, we perform a simple fusion of the two modality images before conducting experiments. The fusion can be formulated as:
> $$
> I_{\text{fused}} = \alpha \cdot I_1 + (1 - \alpha) \cdot I_2.
> $$
> Here, $\alpha = 0.5$, and our experimental results are shown in the table.
>
> | M3FD | MI | VIF | SCD | EN | SSIM | PSNR |
> |-------------|--------|--------|--------|--------|--------|--------
> | Fused image | 3.29 | **0.78**  | **1.68**  | **6.85** | **0.94**  | **63.4**  |
> | Vis | **3.32**  | 0.75 | 1.60  | 6.72 | 0.91 | 63.2 |
>
> | RoadScene | MI | VIF | SCD | EN | SSIM | PSNR |
> |-------------|--------|--------|--------|--------|--------|--------|
> | Fused image | **3.36** | **0.72**  | **1.74**  | **7.38**  | **0.95**  | **62.7** |
> | Vis | 3.22 | 0.67 | 1.66 | 7.28 | 0.93 | 61.8 |
>
> | TNO | MI | VIF | SCD | EN | SSIM | PSNR |
> |-------------|--------|--------|--------|--------|--------|--------|
> | Fused image| **2.91**  | **0.73**  | **1.71**  | **7.12**  | **0.96**  | **62.8** |
> | Vis | 2.68  | 0.68  | 1.61 | 6.99 | 0.91 | 62.6 |
>
> We were pleasantly surprised to find that using the fused image as the sampling starting point led to improved performance compared to using the original visible image, as reflected in the evaluation metrics. This demonstrates that our method still holds
>  **potential for performance gain** when different images are used as starting points. In future versions of the paper, we plan to further explore the fusion performance when using images containing different proportions of modality information as starting points, in order to fully uncover the potential of our method in image fusion.
>
> **Reference**
>
> [1] Zhu Y, Zhao W, Li A, et al. Flowie: Efficient image enhancement via rectified flow[C]//Proceedings of the IEEE/CVF Conference on Computer Vision and Pattern Recognition. 2024: 13-22.
>
> [2] Li J, Cao J, Guo Y, et al. One diffusion step to real-world super-resolution via flow trajectory distillation[J]. arXiv preprint arXiv:2502.01993, 2025.
>
> [3] You W, Zhang M, Zhang L, et al. Consistency Trajectory Matching for One-Step Generative Super-Resolution[J]. arXiv preprint arXiv:2503.20349, 2025.

---

> > ### Author Response · Authors · 2025-08-03
> >
> > We sincerely appreciate the reviewer’s thoughtful feedback and engagement during the discussion. We kindly ask whether the clarifications and additional experiments we have provided have addressed the outstanding concerns. We would be very grateful to hear your thoughts and are happy to provide any further information if needed.

---

> > ### Comment · Reviewer_oayV · 2025-08-04
> >
> > Thank you for the rebuttal. It effectively addressed my concerns, and I have decided to maintain my accept score.

---

### Official Review · Reviewer_fHBh · 2025-07-01

**Clarity:** 3
**Significance:** 4
**Originality:** 4
**Rating:** 5
**Confidence:** 5

**Summary:**

This paper proposes RFfusion, an efficient image fusion method based on rectified flow. RFfusion leverages rectified flow to achieve one-step sampling for image fusion. In addition, it incorporates a variational autoencoder (VAE) to accelerate the fusion process and introduces a two-stage training strategy specifically designed to adapt to the image fusion task. These components work together to significantly enhance both the speed and quality of the fusion results.

**Questions:**

1) The authors mentioned using a pretrained model from rectified flow in their paper. Could the authors provide detailed information about the specifics of this pretrained model?
2) Could the authors provide the performance of the proposed method on additional datasets and other fusion tasks to better demonstrate its generalization capability?
3) The authors employed additional loss functions during VAE training, as shown in Eq. (17). Could the authors explain the rationale behind choosing these specific losses to finetune the VAE?

**Ethical Concerns:**

["NO or VERY MINOR ethics concerns only"]

**Final Justification:**

Previous concerns have been addressed. I decided to keep my original score as Accept.

**Limitations:**

yes

**Quality:**

4

**Strengths And Weaknesses:**

Strengths:
1) Compared to previous methods, the author's proposed approach significantly accelerates image fusion speed and achieves leading performance across multiple tasks.
2) The authors introduced rectified flow and VAE into the image fusion task, achieving promising results and providing inspiration for subsequent research.
3) The authors highlight that their method is adaptable to various fusion tasks and can be transferred to other diffusion-based image fusion approaches, demonstrating exciting generalization potential.

Weaknesses:
1) The authors validate their method exclusively using the baseline DDFM approach. Given that other approaches, such as CCF, also utilize similar image fusion strategies, it would strengthen the paper if the authors could demonstrate the effectiveness of incorporating rectified flow and VAE into these alternative methods to better establish the generalizability of their proposed technique.
2) The authors did not provide performance evaluations of their method on additional image fusion tasks or datasets, such as medical image fusion, thus failing to comprehensively demonstrate the superiority of their approach in broader scenarios.
3) The authors primarily explain why their method can accelerate the original approach; however, according to the results presented in the paper, the proposed method also surpasses DDFM in quantitative metrics. The authors have not provided a detailed explanation for the observed performance improvements.

---

> ### Author Rebuttal · Authors · 2025-07-30
>
> Thank you for the valuable comments. We appreciate the reviewer’s recognition of **the excellent fusion performance** and **generalization ability** of our method. Below are our responses to the reviewer’s questions.
>
> - **W1: Validating the proposed technique on additional fusion methods.**
>
> Thanks for your comments. About weakness 1, you can refer to weakness 2 of **Reviewer uZpR**.
>
> - **W2: The lack of evaluations on additional tasks or datasets.**
>
> We have supplemented our method's performance on medical image fusion. Specifically, we conducted tests on the **Harvard Medical Image Dataset**, which includes **MRI-CT, MRI-PET, and MRI-SPECT** image pairs. The fusion results are shown in the table.
>
> | MRI-CT | MI | AG | SF | EN | SSIM | PSNR |
> |-------------|--------|--------|--------|--------|--------|--------|
> | DDFM [1] | 3.29  | 4.62 | 19.1 | 4.18  | 1.19  | 63.3 |
> | Ours | **3.95**  | **5.49** | **23.6** | **4.62**  | **1.21**  | **64.4**  |
>
> | MRI-PET | MI | AG | SF | EN | SSIM | PSNR |
> |-------------|--------|--------|--------|--------|--------|--------|
> | DDFM [1] | 2.82  | 4.60  | 17.4  | 3.80  | **1.37**  | **65.5** |
> | Ours | **3.11**  | **5.02**  | **19.6** | **3.83**  | 1.01  | 64.8 |
>
> | MRI-SPECT | MI | AG | SF | EN | SSIM | PSNR |
> |-------------|--------|--------|--------|--------|--------|--------|
> | DDFM [1] | 3.04  | **3.77**  | **14.7**  | 3.63  | **1.43**  | 67.4 |
> | Ours | **3.24**  | 3.37 | 14.3 | **3.68**  | 1.33  | **68.1**  |
>
> As shown in the tables, our method consistently outperforms the baseline DDFM on the MRI-CT fusion task, demonstrates a clear advantage on MRI-PET, and achieves comparable performance to DDFM on MRI-SPECT. Overall, our approach exhibits superior performance in medical image fusion compared to DDFM. Notably, our method requires **no additional training** and offers **faster inference**, further highlighting its great potential for medical image fusion applications.
>
> - **W3: Why RFfusion also achieves better quantitative performance than DDFM?**
>
> Our method RFfusion, employs a VAE for **image reconstruction** during the fusion process and is trained on a fusion dataset to better adapt to the fusion task, enabling **high-quality image fusion**. As a result, we achieve improvements in evaluation metrics compared to DDFM. In addition, the pre-trained generative model used in RFfusion provides **better image generation performance** than that of DDFM, which further contributes to the superior performance of our method in terms of quantitative metrics.
>
> - **Q1: Detailed information about the pretrained Rectified Flow.**
>
> We use **checkpoint 12** from the Rectified Flow [2] as the pretrained model, which is trained on the **LSUN Church dataset** with a resolution of **256×256**.
>
> - **Q2: Evaluating the proposed method on additional datasets and fusion tasks.**
>
> Please refer to weakness 2.
>
> - **Q3: The rationale behind the choice of additional loss functions.**
>
> These loss functions are incorporated into the VAE training process to enhance the reconstruction quality of the fused images. They evaluate whether the fused image effectively preserves the critical information from the input images from multiple perspectives, including **structure, color, and saliency**. By introducing diverse loss terms during training, the model not only focuses on the overall image quality but also better retains **the essential information from multiple sources**, thereby improving the comprehensive performance of the fusion results.
>
> **Reference**
>
> [1] Zhao Z, Bai H, Zhu Y, et al. DDFM: denoising diffusion model for multi-modality image fusion[C]//Proceedings of the IEEE/CVF international conference on computer vision. 2023: 8082-8093.
>
> [2] Liu X, Gong C, Liu Q. Flow straight and fast: Learning to generate and transfer data with rectified flow[J]. arXiv preprint arXiv:2209.03003, 2022.

---

> > ### Author Response · Authors · 2025-08-03
> >
> > We sincerely thank the reviewer for the engagement during the discussion. We hope that the additional experiments and clarifications have fully addressed the remaining concerns. If there are any further questions, we would be glad to respond during the remaining discussion period.

---

### Official Review · Reviewer_uZpR · 2025-07-02

**Clarity:** 2
**Significance:** 3
**Originality:** 3
**Rating:** 4
**Confidence:** 4

**Summary:**

This paper proposes a method that leverages Rectified Flow for latent space sampling to accelerate image fusion inference. It introduces VAE to bring the image fusion process into the latent space and employs a two-stage finetuning strategy to adapt the VAE prior to the fusion task, further enhancing inference speed.

**Questions:**

1) The paper mentions that using visible images as the starting point for sampling better results than using Gaussian noise. Please explain the reason behind this choice.
2) In the first stage of VAE training, the authors used the LLVIP and MSRS datasets, while in the second stage only the MSRS dataset was used. Please explain the rationale behind this setup.
3) Please elaborate on the details of the VAE training process, such as the computational cost of the two-stage VAE training, and which type of pre-trained VAE prior is used for finetuning.

**Ethical Concerns:**

["NO or VERY MINOR ethics concerns only"]

**Final Justification:**

The author's answer solved my concerns, so I decided to give it a higher score.

**Limitations:**

see weakness

**Quality:**

3

**Strengths And Weaknesses:**

Strengths:
1) RFfusion successfully addresses the inference time issues of previous diffusion-based methods by significantly accelerating the inference process while maintaining impressive fusion performance.
2) The paper is generally well-structured and coherent, with smooth writing and clear, visually appealing figures.

Weaknesses:
1) Although RFfusion effectively improves inference speed, the quantitative fusion performance presented in the paper does not appear to show a definitive advantage over other methods.
2) RFfusion is only built upon DDFM and has not been validated on other diffusion-based methods, making it difficult to substantiate the generalization capability claimed in the paper.
3) Although RFfusion employs both VAE and Rectified Flow to accelerate inference, the paper lacks ablation studies to separately validate the individual contributions of VAE and Rectified Flow to the overall speed improvement.

---

> ### Author Rebuttal · Authors · 2025-07-30
>
> Thank you for the valuable suggestions. We appreciate the reviewer’s recognition of **the performance improvements** achieved by our method. In response to the questions raised by the reviewer, we have provided the following answers, and we hope they will address the concerns.
>
> - **W1: Quantitative fusion performance lacks a clear advantage over other methods.**
>
> Thanks for your comment. We acknowledge that, compared to other fusion methods, our approach does not demonstrate an absolute and comprehensive advantage across all evaluation metrics. However, in terms of the overall average performance, our method still maintains a leading position. More importantly, rather than merely surpassing existing methods in fusion quality, we aim to emphasize **the significant improvement in fusion speed** achieved by our approach. Compared to the baseline, our method brings substantial enhancements in both **inference speed** and **various performance metrics**. Therefore, we believe that even if more effective diffusion model fusion methods are proposed in the future, our approach can be readily adapted to them and will likely achieve even better results in both speed and effectiveness. This is the core contribution of our work.
>
> - **W2: Lacking validation on other diffusion-based methods.**
>
> We have supplemented our method by integrating it with CCF [1] as the baseline to verify the generalization capability of our approach. The experimental results are shown in the table.
>
> | RoadScene | TE | Qcb | EN | MI | SF | AG | PSNR |
> |-------------|--------|--------|--------|--------|--------|--------|--------|
> | CCF-200 | 6.98  | 0.43 | **7.36** | **2.88**  | 10.1  | 3.88 | **63.3**
> | CCF-1 | **7.33** | **0.45** | 7.01 | 2.31 | **11.3**  | **5.23**  | 58.4
>
> As shown in the table, our method accelerates CCF from **200 steps** sampling to **single step** sampling. Although the performance is somewhat reduced compared to the original CCF method, it still demonstrates the great potential of our method for integration with other diffusion model fusion methods.
>
> - **W3: Lacking ablation studies to isolate the individual contributions of VAE and Rectified Flow to inference speed improvement.**
>
> We have added ablation experiments on the inference speed improvements of VAE and Rectified Flow. The experimental results are as follows, and we will update the article with these results in a later version.
>
> | RF | VAE | Runtime(s)
> |-------------|--------|--------|
> | - | - | 22.03
> | - | ✓ | 0.650
> | ✓ | ✓| **0.308**
>
> The table clearly shows that both VAE and Rectified Flow contribute to **improved inference speed**, which validates the argument presented in our method.
>
> - **Q1: Why use visible images as the sampling starting point?**
>
> Compared to the traditional DDPM algorithm, Rectified Flow focuses more on **direct flow matching between image distributions** and therefore no longer relies on Gaussian noise as a necessary component. By starting the sampling process from a visible image, the model can leverage the visible prior, enabling the generation of more informative and richer fused images.
>
> - **Q2: Only using the MSRS dataset in the second stage.**
>
> In the second stage of VAE training, we introduce the RFfusion sampling process. Since this process relies on inference operations, the training must be conducted with a relatively **small batch size**, and the overall **training time increases significantly**. The primary objective of stage two is to optimize the VAE decoder's ability to **reconstruct fused images**, rather than to focus on the generalization performance of the fusion method across multiple datasets. Therefore, we choose to train on a relatively small dataset, which significantly reduces the training time without compromising the final performance of our method.
>
> -**Q3: The paper should clarify the VAE training details.**
>
> In the first stage of training, we trained for a total of **20 epochs** on the LLVIP [2] and MSRS [3] datasets. However, we observed that the model typically converged within **4 to 5 epochs**, which took approximately **4 to 5 hours** on a V100 GPU. In the second stage, we trained for **40 epochs** on the MSRS dataset, with the entire process taking about **one day**. The pre-trained VAE we used was the VQ-VAE from LDM [4], with a **downsampling factor of f = 4** and equipped with an **attention mechanism**.
>
> Reference
>
> [1] Cao B, Xu X, Zhu P, et al. Conditional controllable image fusion[J]. Advances in Neural Information Processing Systems, 2024, 37: 120311-120335.
>
> [2] Jia X, Zhu C, Li M, et al. LLVIP: A visible-infrared paired dataset for low-light vision[C]//Proceedings of the IEEE/CVF international conference on computer vision. 2021: 3496-3504.
>
> [3] Tang L, Yuan J, Ma J. Image fusion in the loop of high-level vision tasks: A semantic-aware real-time infrared and visible image fusion network[J]. Information Fusion, 2022, 82: 28
>
> [4] Rombach R, Blattmann A, Lorenz D, et al. High-resolution image synthesis with latent diffusion models[C]//Proceedings of the IEEE/CVF conference on computer vision and pattern recognition. 2022: 10684-10

---

> > ### Author Response · Authors · 2025-08-03
> >
> > We thank the reviewer for engaging in the discussion. We are eager to know whether the additional experiments and clarifications we provided have addressed the reviewer’s concerns. If there are any further questions during the remaining time of the discussion phase, we would be more than happy to respond.

---

> > > ### Comment · Reviewer_uZpR · 2025-08-03
> > >
> > > The author's answer solved my concerns, so I decided to give it a higher rating.

---

### Note · Authors · 2025-08-13

We would like to extend our sincere gratitude to each reviewer for their insightful feedback. All four reviewers recognized the merits of our paper and rated it as either **borderline accept** or **accept**. We are delighted to know that you value our contributions, and we greatly appreciate your comments highlighting the strengths of our work:

- Our paper presents a clear logical flow and a well-organized structure.  ( Reviewer **uZpR**, **eb1j** )

- The proposed method greatly accelerates inference speed.  ( Reviewer **uZpR**, **fHBh**, **oayV** )

- Our method demonstrates strong generalization across diverse fusion tasks and methods. ( Reviewer **fHBh** )

- Technically, our method is novel and offers valuable insights for future fusion work. ( Reviewer **fHBh**, **eb1j** )

To address the reviewers’ concerns regarding **generalization**, **ablation studies**, and **model architecture**, we conducted additional experiments and provided further explanations. Following our responses, the reviewers indicated that their concerns had been resolved. Specifically:

- We added experiments using the CCF method to verify the generalization ability of our approach.

- We performed additional ablation studies on method design and inference speed.

- We included experiments on medical image fusion to further validate generalization.

- We explained the design rationale of our method and the underlying principles that enable effective fusion.

Finally, we wish to express our heartfelt thanks to the reviewers for their careful evaluation and constructive feedback.

---

### Decision · Program_Chairs · 2025-09-17

**Decision:**

Accept (poster)

**Comment:**

This paper proposes RFfusion, a novel framework that integrates Rectified Flow and task-specific VAEs for efficient image fusion. The core contribution lies in enabling one-step latent space sampling, combined with a two-stage VAE training strategy, which significantly accelerates inference while maintaining high-quality outputs. Strengths include clear methodological design, strong empirical performance across multiple fusion tasks (infrared-visible, multi-exposure, multi-focus), and inspiring generalization potential for diffusion-based methods. Weaknesses noted by reviewers include limited validation on broader datasets, lack of ablations for individual components, and insufficient exploration of modality-specific representations. However, the rebuttal provided clarifications on VAE training choices and explained fusion dynamics, alleviating major concerns. While additional experiments on medical fusion or other backbones would further strengthen the work, the proposed approach makes a meaningful and timely contribution by demonstrating both speedup and quality gains. Balancing innovation, practical utility, and solid evaluation, I recommend acceptance.